# Preoperative EUS vs. PET-CT Evaluation of Response to Neoadjuvant Therapy for Esophagogastric Cancer and Its Correlation with Survival

**DOI:** 10.3390/cancers15112941

**Published:** 2023-05-27

**Authors:** Victor Amezcua-Hernandez, Rita Jimenez-Rosales, Juan Gabriel Martinez-Cara, Javier Garcia-Garcia, Francisco Valverde Lopez, Eduardo Redondo-Cerezo

**Affiliations:** 1Department of Oncology, “Virgen de las Nieves” University Hospital, 18014 Granada, Spain; 2Department of Gastroenterology, “Virgen de las Nieves” University Hospital, 18014 Granada, Spain; 3Instituto de Investigación Biosanitaria de Granada (ibs.GRANADA), 18014 Granada, Spain

**Keywords:** esophagogastric junction adenocarcinoma, gastric adenocarcinoma, EUS, PET-CT, neoadjuvant therapy, prognosis

## Abstract

**Simple Summary:**

Endoscopic ultrasonography is commonly used for the initial staging of esophago-gastric adenocarcinoma. However, the preoperative staging after neoadjuvant therapy remains controversial, with PET-CT and endoscopic ultrasonography suggested but not universally applied. In this study, we examined a large series of gastric and esophago-gastric junction adenocarcinomas, where both procedures were performed as part of the initial staging and after preoperative chemotherapy. Our findings indicate that both EUS and PET-CT have limitations but can determine the stage of cancer and predict survival. Notably, the study showed that the assessment of lymph nodes using endoscopic ultrasonography and the evaluation of response to preoperative chemotherapy were predictive of survival, underscoring the importance of this procedure as an additional tool for staging in this particular setting.

**Abstract:**

Background: The objective of our study was to investigate whether Endoscopic Ultrasonography (EUS) and Positron Emission Tomography-Computed Tomography (PET-CT) restaging can predict survival in upper gastrointestinal tract adenocarcinomas and to assess their accuracy when compared to pathology. Methods: We conducted a retrospective study on all patients who underwent EUS for staging of gastric or esophago-gastric junction adenocarcinoma between 2010 and 2021. EUS and PET-CT were performed, and preoperative TNM restaging was conducted using both procedures within 21 days prior to surgery. Disease-free survival (DFS) and overall survival (OS) were evaluated. Results: A total of 185 patients (74.7% male) were included in the study. The accuracy of EUS for distinguishing between T1-T2 and T3-T4 tumors after neoadjuvant therapy was 66.7% (95% CI: 50.3–77.8%), and for N staging, the accuracy was 70.8% (95% CI: 51.8–81.8%). Regarding PET-CT, the accuracy for N positivity was 60.4% (95% CI: 46.3–73%). Kaplan–Meier analysis revealed a significant correlation between positive lymph nodes on restaging EUS and PET-CT with DFS. Multivariate COX regression analysis identified N restaging with EUS and PET-CT, as well as the Charlson comorbidity index, as correlated factors with DFS. Positive lymph nodes on EUS and PET-CT were predictors of OS. In multivariate Cox regression analysis, the independent risk factors for OS were found to be the Charlson comorbidity index, T response by EUS, and male sex. Conclusion: Both EUS and PET-CT are valuable tools for determining the preoperative stage of esophago-gastric cancer. Both techniques can predict survival, with preoperative N staging and response to neoadjuvant therapy assessed by EUS being the main predictors.

## 1. Introduction

Adenocarcinomas of the stomach (GC) and esophagogastric junction (AEG) (Types I-III according to the Sievert classification) [1] are among the gastrointestinal tumors with the highest mortality worldwide [2]. Accurate staging plays a crucial role in the treatment of gastric cancer patients, as prognosis is determined by tumor growth, including lymph node involvement and the extent of disease to neighboring organs [3]. Therefore, precise staging is essential prior to making any decisions regarding definitive treatment, as inaccuracies in staging have been shown to negatively impact patient outcomes [4].

However, there is a lack of data on the actual predictive ability of endoscopic ultrasonography (EUS) staging of esophageal and gastric adenocarcinomas in real-life settings. Comparisons with post-resection histology can vary due to different neoadjuvant therapy (NT) regimens, and pathologic inaccuracies may also exist.

Furthermore, while there is an ongoing debate regarding the role of EUS in tumor staging after neoadjuvant therapy, as it is not recommended by worldwide guidelines, some authors have recognized its value, and it is routinely performed in many centers [5,6].

Current guidelines for staging gastric, esophageal, and esophagogastric neoplasms are quite similar, recommending abdominal computed tomography (CT) with oral and intravenous contrast and considering Positron Emission Tomography-Computed Tomography (PET-CT) in cases of suspected metastatic disease. EUS is suggested for local assessment of esophageal layers and node involvement [7,8]. However, depending on availability, PET-CT with oral and intravenous contrast may be sufficient for comprehensive restaging of these cancers, reducing the need for additional diagnostic procedures and improving the detection of metastatic disease.

Considering the controversy surrounding the role of EUS in restaging esophageal and gastric adenocarcinoma, the objective of our retrospective study was to evaluate survival outcomes based on T and N changes after neoadjuvant therapy, as assessed by EUS and PET-CT. As secondary endpoints, we analyzed the accuracy of EUS and PET-CT compared to pathological results, as well as the key factors in the pretreatment workup that are associated with survival in these patients.

## 2. Materials and Methods

### 2.1. Study Design

This retrospective single-center study utilized a prospectively collected database of patients who underwent EUS for cancer staging between January 2010 and January 2021. The inclusion criteria consisted of patients who underwent surgical resection and were diagnosed with gastric or esophagogastric adenocarcinoma. Patients who had undergone endoscopic mucosal resection (EMR), had widespread metastatic disease, were unfit for surgery or neoadjuvant therapy, or did not undergo a complete staging workup were excluded. The patients underwent an initial baseline assessment followed by preoperative TNM staging using EUS and PET-CT, performed two weeks after completing neoadjuvant chemotherapy and within 21 days prior to the surgical treatment, in accordance with the guidelines of the National Comprehensive Cancer Network^®^ (NCCN^®^, Plymouth Meeting, PA, USA) [7,8]. The neoadjuvant therapy primarily consisted of ECF (epirubicin, cisplatin, and fluorouracil), FLOT-4 (fluorouracil plus leucovorin, oxaliplatin, and docetaxel), or EOX (epirubicin, capecitabine, and oxaliplatin) schedules.

### 2.2. Data Collection

We collected comprehensive information on patients’ general demographic features, tumor characteristics, and pre-existing conditions, including comorbidities and Charlson and ECOG scores. Staging results obtained from both EUS and PET-CT before and after neoadjuvant therapy was recorded. Surgical procedures and types of lymphadenectomy performed were documented. Pathologic results after surgery, including tumor stage and lymph node involvement, were noted. Additionally, information on the type and location of recurrences as well as patient mortality was collected (see Table 1). EUS was only performed if there was no metastatic disease.

EUS response to neoadjuvant therapy was objectively determined based on the T and N staging before and after treatment. Responses were classified as either “response” (indicating a response in either T or N stage with stability or response in the other), “stable disease” (no change in T and N stage before and after therapy), or “progression” (progression in either T or N stage or both).

We also included PEC-CT staging and at least N and M classification. Survival information was obtained from electronic medical charts and supplemented with telephone calls if necessary. The regional electronic medical system integrates information from hospital and primary care settings, allowing for comprehensive patient tracking and close follow-up. When patients did not die while admitted to hospital, information regarding patient deaths was obtained from family physicians records on the system and death certificates. In cases of unclear information, direct contact was made with patients or their household relatives via telephone calls. The only patients lost to follow-up were those who had died.

### 2.3. EUS Staging

EUS procedures were conducted by two highly experienced endoscopists, E.R.-C. and J.G.M.-C., who had 7 and 5 years of EUS experience at the beginning of the study, respectively. Both endoscopists performed more than 350 procedures per year. The procedures were performed under nurse-based propofol sedation using a radial EUS probe (Olympus, GF-UCT 165-AL5, Olympus, Tokyo, Japan). EUS examinations were carried out before and after neoadjuvant treatment. A systematic and comprehensive evaluation of the gastrointestinal tract was performed during each EUS procedure. The examination was initiated in the descending portion of the duodenum, with particular attention given to the aorto-caval region, except in patients with gastric outlet obstruction. Stomach inspection was typically conducted during endoscope retrieval, following instillation of water and with the assistance of a balloon in the transducer tip. Every structure and lymph node was carefully studied, including the mediastinal and upper abdominal lymph nodes. Local tumor infiltration was assessed using the five-layer structure of the gastric wall. The N stage assessment in EUS was based on the number of metastatic perigastric lymph nodes. Lymph node metastasis was determined based on the presence of two or more of the following criteria: size greater than 5 mm, round shape, hypoechoic pattern, and smooth border. A comprehensive examination of the pancreas, hepatic hilum, stomach, and mediastinum was performed in every case. Fine needle aspiration or biopsies were not routinely performed. Instead, lymph nodes were considered positive if they exhibited previously described suspicious characteristics (hypoechoic, sharply demarcated borders, or rounded contour) and had a diameter of more than 0.5 cm [9,10,11]. This practice is commonly followed in different settings due to the difficulty of sampling lymph nodes without puncturing the main tumor, thus providing additional information only in a few cases. Staging was performed according to the TNM classification in every EUS case.

### 2.4. PET-CT Staging

All patients underwent a minimum fasting period of 4 h prior to the FDG-PET-CT study. Before the administration of FDG, the blood glucose concentration was measured, and the study proceeded only if the glucose concentration was below 200 mg/dL. All FDG-PET-CT imaging was performed using a hybrid PET-CT scanner (Biograph LSO 2; Siemens Medical Solutions, Malvern, PA, USA). The CT component of the PET-CT studies was conducted without the administration of an intravenous contrast agent. CT images with 5 mm slices were acquired from the base of the skull through the proximal thighs using 130 kVp and 110 mA. Emission PET-CT images were obtained over the same anatomical range starting 45 to 60 min after the administration of 555 to 740 MBq FDG (15 to 20 mCi FDG), with imaging times ranging from 2 to 4 min per bed position, depending on patient weight. PET-CT images were carefully reviewed to identify abnormal FDG uptake at the primary tumor site, lymph node regions, and distant sites. In the PET-CT scans, the presence of metastases, celiac lymph node involvement, other affected lymph chains, and maximum standardized uptake value (SUVmax) before and after neoadjuvant treatment were recorded. The FDG PET-CT scans were primarily directed towards N and M staging, while T staging was determined by EUS. Lymph nodes were considered positive for metastasis when there was 18F-FDG uptake higher than that in the liver. Other factors such as lymph node short axis and SUVmax were also considered when studying PET/CT N staging.

### 2.5. Follow Up

All patients were closely monitored during the follow-up period. They underwent imaging evaluations using CT scan or PET-CT, as well as regular clinical assessments every 3 months for a duration of three years and then every 6 months thereafter. The objective of these evaluations was to detect any signs of tumor relapse and assess the type of relapse. Additionally, the occurrence of death was recorded. Overall survival was calculated from the time of diagnosis to the time of death. In cases where patients were lost to follow-up, we were able to access and review their primary care charts through the integrated clinical charts of the healthcare service. This ensured that we could gather as much information as possible, even for patients who were not actively participating in the follow-up.

### 2.6. Outcomes

The main outcome of the study was to assess disease-specific survival in different T and N restaging groups as determined by EUS and PET-CT, comparing them with the histological reports. For EUS staging, a subgroup analysis was conducted to compare T1-T2 tumors with T3-T4 tumors. This comparison was based on the fact that current guidelines recommend neoadjuvant therapy for T3-T4 tumors, indicating a more advanced stage of the disease [7,8].

A secondary outcome of the study was to evaluate the presence of small amounts of ascites, which may not be detectable by MDCT and PET-CT, during the initial EUS examination. The influence of ascites on survival was investigated as an indicator of an extended disease.

### 2.7. Statistical Analysis

The statistical analysis included univariate analysis using the Chi-square test to assess the accuracy of EUS and PET-CT staging compared to the pathological results. Disease-free survival (DFS) and overall survival (OS) were analyzed using Kaplan–Meier survival curves. Cox regression analysis was employed to control for confounding factors such as patients’ previous conditions and assess their influence on survival.

The statistical analysis was conducted using SPSS 25 software (IBM Corp., Armonk, NY, USA). A significance level of *p* < 0.05 was considered statistically significant.

### 2.8. Ethics

This retrospective analysis was conducted on a prospectively collected database, which had received approval from the institution’s Human Research Committee in 2011 (Comité provincial de ética de Granada. Approval code: GR2011.23. Date: 22 February 2011). Prior to the procedures, all patients provided informed consent for the treatment and inclusion of their data in the study database. The study was conducted in compliance with the ethical standards outlined by the committee on human experimentation at both the institutional and national levels. Furthermore, the study adhered to the principles set forth in the Helsinki Declaration of 1964 and subsequent revisions.

## 3. Results

One-hundred and eighty-five patients with gastric or esophago-gastric adenocarcinoma submitted for EUS staging were included, of which 139 were male. Mean age was 66.7 years (range 25–89 years). Their main symptom was dysphagia (31.7%), followed by hematemesis (22%), constitutional syndrome (16.1%), and abdominal or chest pain (15.6%) (See Table 1). Median follow up was 23 months (range 7.5–130 months)

### 3.1. Tumor Characteristics

Most of the tumors were gastric adenocarcinoma (69.4%), located in the gastric body (40.3%), antrum (25.3%), and esophagogastric junction (33,4%). Endoscopically, we found a significant stenosis in 32.8%. EUS TNM staging and restaging is depicted in Table 1. Small quantities of ascites, undetected by other imaging methods, were found in 28 patients (16.2%). PET-CT found positive lymph nodes in 38.2% of patients, whereas EUS found positive lymph nodes in 71.3% on the initial staging. After neoadjuvant therapy, EUS response was observed in 54% of patients. The mean SUVmax value on PET-CT imaging for the primary tumor prior to chemotherapy showed a significant decrease after the treatment (9.96 ± 6.6 vs. 6.7 ± 4.6; *p* < 0.0001). We found metastatic spread on PET-CT in 12% of patients on baseline assessment and in 15% after preoperative therapy.

### 3.2. Neoadjuvant Therapy

All patients included in the study underwent perioperative chemotherapy with established standard treatment regimens prior to surgery. None of the patients received radiotherapy as it is typically reserved for adjuvant therapy in high-risk patients who did not undergo neoadjuvant or perioperative therapy. The main chemotherapy regimen administered to the patients was ECF, which was given to 56% of the patients. The FLOT 4 regimen was administered to 30% of the patients, while the remaining 14% received the EOX regimen. Refusal of chemotherapy was considered an exclusion criterion.

### 3.3. Surgical Treatment

Among the patients who received surgical treatment, 40.9% underwent total gastrectomy; 34.9% underwent transhiatal esophagectomy, and 24.2% underwent partial gastrectomy.

### 3.4. Accuracy of EUS and PET-CT in Restaging

EUS overall accuracy for the distinction of T1-T2 vs. T3-T4 tumors after neoadjuvant therapy was 66.7% (95% CI 50.3–77.8%; kappa: 0.17). When considering the presence of positive lymph nodes, EUS showed an accuracy of 70.8% (95% CI: 51.8–81.8%; kappa 0.39) (Figure 1).

Accuracy for T1 was 92.4% (95% CI: 87.7–95.4%) accuracy for T2 was 70.4% (95% CI 63.3–76.6%); accuracy for T3 was 54.2% (95% CI: 46.9–63.3%); accuracy for T4 was 72.6% (95% CI: 65.7–78.6%). Regarding N stage accuracy, for N0 it was 71.4% (95% CI: 64.5–77.4%); for N1 it was 72.4% (95% CI: 65.6–78.4%); for N2 it was 83.2% (95% CI: 77.2–87.9%), and for N3 it was 94% (95% CI: 89.7–96.6%). The main tendency when mistaking T staging was over-staging; by contrast, except for N0 tumors, the main tendency in N staging when EUS was not accurate was a down-staging.

Regarding PET-CT, we found that N positivity showed an overall accuracy of 60.4% (95% CI: 46.3–73%, kappa 0.16). No correlation was found between SUVmax in the tumor and histological stage (Figure 2).

Sensitivities, specificities, and positive and negative predictive values are shown in Table 2.

### 3.5. Survival Analysis

Mean overall survival (OS) was 45 months (95% CI: 38–55 months) with a 5 year survival rate of 25.7%. The median disease-free survival (DFS) was 38 months (95% CI: 32–44 months). Seventy-eight patients suffered metastatic recurrence (45,6%), and twelve a local recurrence (7%). The most common metastatic site was the peritoneum, followed by the liver.

### 3.6. Disease-Free Survival

In the Kaplan–Meier analysis, we observed a significant correlation between positive lymph nodes in preoperative EUS (yUN) and restaging PET-CT with disease-free survival (DFS) (Figure 3 and Figure 4).

Furthermore, the pathologic T stage, even when grouped in T1-T2 vs. T3-T4, did not reach significant differences in DFS. In multivariate COX regression analysis, we found that N restaging with EUS and PET-CT was significantly correlated with DFS, as well as the Charlson comorbidity score (Table 3).

### 3.7. Overall Survival

EUS restaging of positive lymph nodes was also a predictor for OS, as well as PET-CT positive nodes. As with DFS, positive nodes found in the pathologic analysis were also related to OS. The pathologic T stage behaved similarly with OS as with DFS, with no significant differences even when grouped in T0-T2 stages vs. T3-T4 stages. In the multivariate Cox regression analysis-only Charlson Score, the T response by EUS and male sex were independent risks factors for OS (Table 4).

No differences were found either in OS or in DFS between the different chemotherapy schemes. No significant differences were observed in EUS or PET-CT prediction ability between them.

As an independent analysis, but closely related to T staging, the finding of small pools of perigastric ascites was closely related with overall survival, with significantly longer survival for patients without those small ascites pools (Figure 5). However, it was not significant as an independent predictor for survival in the Cox regression analysis. 

## 4. Discussion

Restaging using EUS remains a topic of debate in upper GI tract cancers and is not currently included in the institutional guidelines followed by most oncologists for post-neoadjuvant therapy work-up [7,8]. However, several research papers have demonstrated the potential of EUS in restaging these patients [6,12,13,14,15]. In this study, we conducted a comprehensive evaluation of the restaging accuracy of both EUS and PET-CT, as well as their predictive ability for patient survival. Our results indicate that neither of these diagnostic procedures are perfectly accurate for restaging, but contrary to previous assumptions, EUS is at least as accurate as PET-CT in preoperative staging. Furthermore, we observed a correlation between EUS and PET-CT restaging and disease-free survival, a finding that was consistent when examining overall survival, as expected. Similar findings have been reported in previous smaller studies that did not specifically differentiate between preoperative EUS N and T evaluation and did not include PET-CT [6,16,17].

Neoadjuvant chemotherapy has been shown to improve surgical outcomes in patients with gastric and esophagogastric junction adenocarcinomas [18]. In real-world clinical practice, the availability of reliable tools for preoperative evaluation is crucial. These tools should provide valuable information for clinical decision-making and be capable of predicting survival outcomes. However, a CT scan alone has not demonstrated reliable accuracy in evaluating tumor response to chemotherapy [19]. Therefore, a multimodal approach incorporating multiple diagnostic tools is warranted. In fact, down-staging following neoadjuvant treatment has been recognized as a strong independent predictor of survival, even after adjusting for factors such as patient age, tumor grade, pre-treatment stage, lymphovascular invasion, resection margin status, and surgical resection type [20].

Our findings revealed that the accuracy of EUS following neoadjuvant therapy was 66.3% for differentiating between T1-T2 and T3-T4 tumors, with the lowest accuracy observed for T3 tumors (54.2%). In terms of N staging, we achieved over 70% accuracy across all stages. While these results are not excellent, they demonstrate similar precision to previous studies. The sensitivity/specificity for N staging was 60%/78.6% for EUS and 27.3%/88.5% for PET-CT, with EUS performing better in N restaging [6,21]. In our clinical practice, fine-needle aspiration (FNA) for lymph node sampling is rarely performed. This is mainly due to the difficulty of performing FNA without piercing the primary tumor and, in other cases, because the cytology results would not significantly alter the overall disease management and treatment plan. Nonetheless, EUS exhibited superior performance compared to PET-CT in N staging.

EUS T staging demonstrated accuracies ranging from 54% to 72% but did not have a significant predictive value for survival. However, evidence of response on EUS after neoadjuvant therapy was identified as an independent risk factor for OS. It is important to note that the challenges associated with tumor restaging, such as inflammatory response and layer artifacts induced by tumor necrosis, make EUS suboptimal for this purpose. Previous studies have indicated that changes in layer structure, which are crucial for initial staging, play a lesser role in restaging, where tumor size becomes the primary objective measure associated with pathological response [17,22]. In our clinical practice, a reduction in tumor maximal thickness, improvement of a previous stenosis, and the subjective impression of the endoscopist are considered the best predictors of response and are considered accordingly.

Although EUS has shown only moderate accuracy for T staging after neoadjuvant therapy, N staging on EUS has demonstrated its predictive value for survival, with T staging identified as an independent risk factor for OS. The importance of N staging as a major prognostic factor for survival has been previously recognized, and our EUS and PET-CT series align with these findings [23]. To enhance the ability of EUS for T staging, further strategies are required to address any uncertainties surrounding the broader implementation of this procedure in the perioperative workup of these patients.

In our study, we identified small perigastric ascites pools, measuring less than 1 cm in diameter, in 16% of patients. The endoscopist evaluated these pools as indicative of T4a disease, suggesting infiltration of the visceral peritoneum by the tumor [24,25]. Interestingly, we found significant differences in survival between patients who lacked these small ascites pools and those who had them. These low-volume ascites pools, which often go unnoticed on imaging modalities such as a CT scan, PET-CT, and even laparoscopy, have an important impact on predicting patients’ survival. They signify the presence of advanced disease, as previously described in the initial staging [26], but their significance in restaging has not been widely recognized before.

Our overall survival results may appear disappointing when compared to certain studies, as they are nearly half of the reported figures in some papers [6]. However, they are more consistent with the findings of other studies [27]. This discrepancy can be attributed to the relatively advanced stage of disease at the time of diagnosis in our series, with a majority of patients presenting with T4 stage tumors. Despite this, the surgical outcomes were favorable, with a high proportion of patients achieving R0 resections. It is important to note that 70% of patients diagnosed with an esophagogastric tumor underwent surgical treatment, indicating the potential effectiveness of surgery in our patient population.

Our study has several limitations that should be acknowledged. Firstly, it is a retrospective analysis conducted at a single center, which may introduce inherent biases despite the prospective collection of data. A dedicated prospective study involving multiple centers could provide more robust evidence. However, the advantage of our study lies in the uniformity of EUS performance, as it was conducted by only two highly experienced endosonographers. Secondly, patients with metastatic disease were excluded from undergoing EUS evaluation. This selection criteria may have led to the inclusion of patients with less advanced disease. However, we aimed to reflect real-life clinical practice and adhere to the standard management protocols followed in our center.

## 5. Conclusions

In conclusion, despite their limitations, EUS and PET-CT remain valuable tools in the preoperative evaluation of gastro-esophageal adenocarcinoma. While their accuracy may not be perfect, they provide important information that aids in clinical decision-making. PET-CT can complement EUS information in the detection of distant disease and peritoneal involvement, confirmation of malignancy of incidental findings on examination (i.e., ascites lakes), and confirmation of malignancy of suspicious lymph nodes detected on EUS. It can also increase the cost-effectiveness of cytology during the endoscopic procedure, directing efforts to the lymph nodes with the highest uptake of radiotracer, therefore more profitable, and can be used in pre-surgical planning. However, it is worth noting that emerging technologies such as gallium 68 (68Ga)-labeled fibroblast-activation protein inhibitors (FAPIs) or PET-MRI hold promise for potentially improving the diagnostic accuracy in the future. Further research and advancements in imaging modalities are needed to enhance the precision of preoperative staging and ultimately improve patient outcomes in this challenging disease.

## Figures and Tables

**Figure 1 cancers-15-02941-f001:**
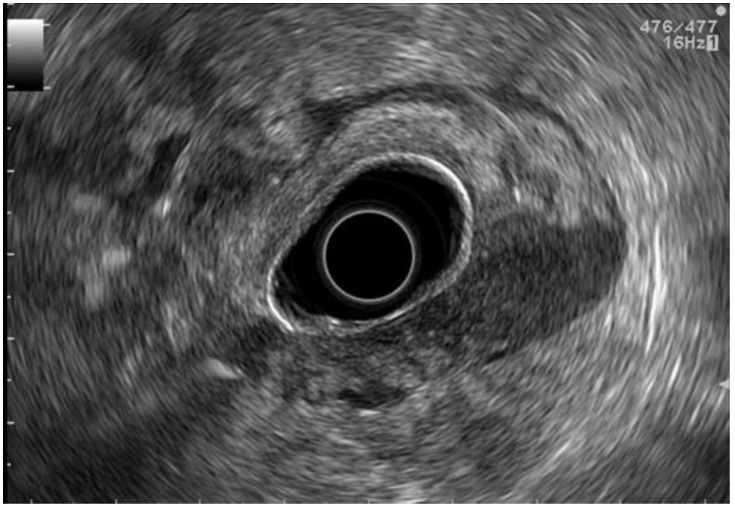
EUS staging. Siewert III adenocarcinoma with a small quantity of perigastric ascites.

**Figure 2 cancers-15-02941-f002:**
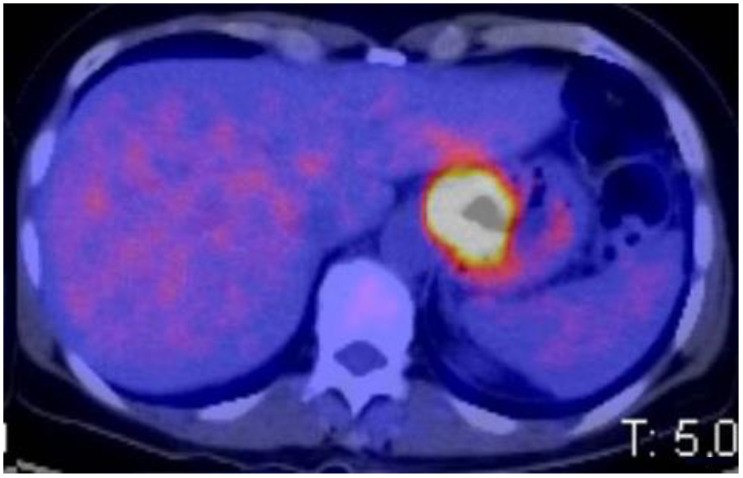
PET-CT: Siewert III adenocarcinoma.

**Figure 3 cancers-15-02941-f003:**
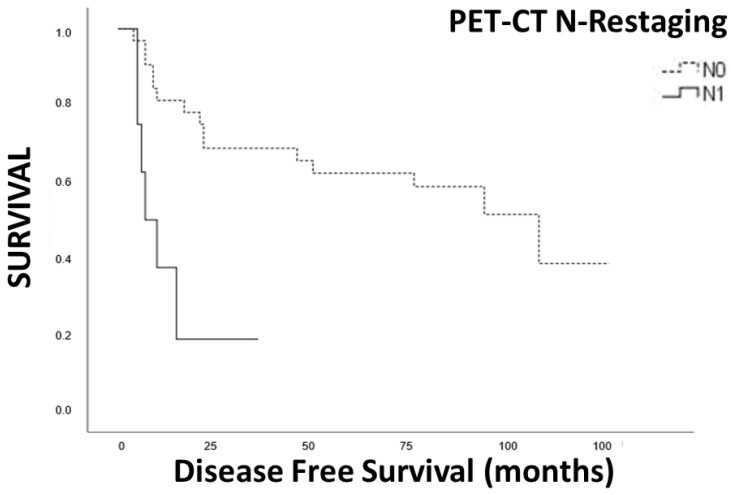
Survival analysis N positive vs. N negative EUS restaging.

**Figure 4 cancers-15-02941-f004:**
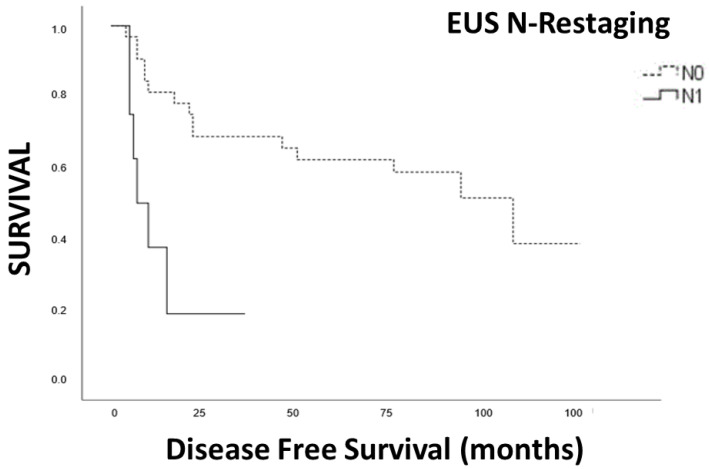
Survival analysis N positive vs. N negative PET-CT restaging.

**Figure 5 cancers-15-02941-f005:**
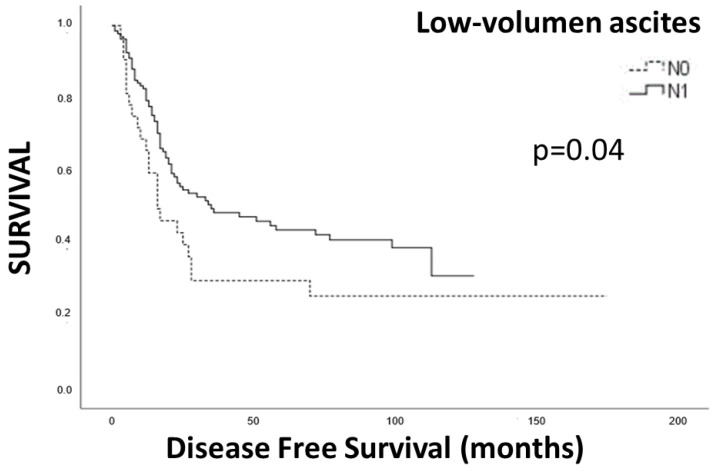
Survival analysis for low volume ascites.

**Table 1 cancers-15-02941-t001:** Patients’ general characteristics.

Parameter	*n*	%
Age (median. Years)	67	
Gender		
Male	139	62.5
Female	46	21.4
Localization		
Distal esophagus	15	8.1
Gastroesophageal junction	41	22
Upper stomach	129	69.4
Grading		
G1	9	4.9
G2	104	56.5
G3	71	38.6
Lauren’s classification		
Intestinal	97	52.4
Mixed	15	8.3
Diffuse	73	39.3
**Staging**		
Depth of invasion (EUS)		
T1	2	1.1
T2	8	4.5
T3	54	30.2
T4	117	65.4
Node detected (EUS)		
N0	49	28.7
N1	48	28.1
N2	55	32.2
N3	19	11.1
Node detected (PET-CT)		
N (+)	58	38.2
N (−)	94	61.8
**Restaging**		
Depth of invasión		
T1	1	0.54%
T2	44	23.91
T3	65	35.4
T4	74	40
Node detected (EUS)		
N0	96	52
N1	43	23.2
N2	43	23.2
N3	2	1.4
Node detected (PET-CT)		
N (+)	49	26.7
N (−)	136	73.3
Histopathologic T category (ypT)		
ypT0	20	11.4
ypT1	15	7.9
ypT2	29	15.8
ypT3	80	43
ypT4	41	21.9
Histopathologic N category (ypN)		
ypN0	96	51.8
ypN1	41	21.9
ypN2	27	14.9
ypN3	21	11.4

**Table 2 cancers-15-02941-t002:** Malignant lymph nodes detection in restaging.

	EUS	PET-CT
	% [95% CI]	% [95% CI]
Sensitivity	60 [38.7–78.1]	27.3 [13.2–48.2]
Specificity	78.6 [60.5–89.8]	88.5 [71–96]
PPV	73.3 [55.6–85.8]	59.0 [43.4–72.9]
NPV	66.7 [43.7–83.7]	66.7 [35.4–87.9]
Accuracy	70.8 [1.54–19.6]	60.4 [46.3–73]

**Table 3 cancers-15-02941-t003:** Survival analysis for DFS.

	*n*	Median (CI 95%) (Months)	*p*
EUS restaging			
N+	98	22 (15.47–28.53)	0.001
N0	72	108 (26.34–189.65)	
PET-CT restaging			
N+	34	13.69 (5.25–22.11)	0.001
N0	136	79.13 (61.44–96.81)	

**Table 4 cancers-15-02941-t004:** Multivariate analysis.

Disease Free Survival (DFS)			
	Hazard Ratio	95% CI	*p*
Age	0.95	0.89–1.01	0.12
Sex	0.768	0.20–2.93	0.69
Charlson	2.51	1.25–5.04	0.009
N+ restaging PET-CT	20.91	3.39–129.08	0.001
N+ restaging EUS	4.37	1.09–17.54	0.037
N+ pathology	4.68	0.92–23.91	0.063
**Overall Survival (OS)**			
	**Hazard Ratio**	**95% CI**	** *p* **
Age	1.004	0.93–1.08	0.29
Sex (female)	0.24	1.09–3.09	0.12
Charlson	1.83	1.21–6.57	0.023
N response to NT on EUS	3.02	0.88–11.62	0.145
T response to NT on EUS	0.09	0.02–0.46	0.004
N+ restaging EUS	0.85	0.22–3.21	0.44
N+ restaging PET-CT	0.34	0.08–1.40	0.20
N+ pathology	10.72	1.56–77.77	0.016
Low volumen ascites	1.374	0.51–3.68	0.9

## Data Availability

Full study protocol can be requested from the corresponding author. Dataset should be shared as reasonably requested.

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
