# Peer review of "Preoperative EUS vs. PET-CT Evaluation of Response to Neoadjuvant Therapy for Esophagogastric Cancer and Its Correlation with Survival"

_cancers, 2023, doi:10.3390/cancers15112941_

Round 1

Reviewer 1 Report

  1. Author selected only one institution for their study instead of multiple.
  2. Should also consider FNAC and histopathology for TNM grading.
  3. Author should depict images of EUS and PET-CT in the paper.
  4. Which neo adjuvant therapies were used , is not discussed 
  5. A compititve images of PET-CT images before and after chemotherapy would be advantageous 
  6. any Immune correlation viz-a viz TNM staging would also benefit readers of this study 

Author Response

  • Author selected only one institution for their study instead of multiple.

Indeed. This is a long-term study addressing a procedure that is quite operator dependent. Being a real-life study, I’m sure a multicenter study would have provided a different perspective, but we believed reliability was better warranted with this design, given the possible bias provoked by the inclusion of multiple endoscopists. However, to some extent we agree with you. Thanks for the comment.

  • Should also consider FNAC and histopathology for TNM grading.

We did, many times it is difficult to sample a lymph node without traversing the tumor. This can be so common that FNA becomes a quite infrequent added tool for staging. We agree with you this need to be explained, and a statement has been added to the article, lines 116-119.

  • Author should depict images of EUS and PET-CT in the paper.

Images have been added conveniently.

  • Which neo adjuvant therapies were used , is not discussed 

We understand the importance of chemotherapy in this study, but we did not intend to focus on the influence of the different schemes on results, which have interest, but on the endoscopic and radiological workup itself. However, we have added a small sentence, to clarify this aspect, in lines 192-196. Thank you for the interesting remarks.

  • A compititve images of PET-CT images before and after chemotherapy would be advantageous 

We have added some images. They can have an obvious visual interest for the paper, but it is also limited

  • any Immune correlation viz-a viz TNM staging would also benefit readers of this study 

Sorry, I do not fully understand this statement.

An in-depth revision of the English language has been performed.

Reviewer 2 Report

Thank you very much for allowing me to review this manuscript. The authors describe a retrospective study evaluating the performance of endoscopic ultrasound (EUS) and PET/CT for the restaging of esophageal, esophagogastric junction, and gastric cancer and correlate this restaging to survival metrics. The manuscript's strengths include a moderately large sample and sound statistical methods to determine survival. Weaknesses include the commonality of the approach, expected results, and suboptimal narrative structure. EUS is a mainstay of the locoregional staging of these cancers. FDG PET-CT has many limitations that are being overcome with the rise of new radiopharmaceuticals such as FAPI and other imaging technologies such as hybrid PET-MRI. Therefore, this manuscript, unfortunately, carries the burden of using out-of-date methods. 

I raise below some points that could be better explained or improved:

Abstract:

Line 11 - EUS and PET-CT are used in the abbreviated form without a prior introduction

Line 24 - "EUS and PET-CT are suboptimal tools, but the most accurate, to determine esophagogastric cancer stage" - I would advise rephrasing this sentence as it can be deceiving. Optimal means the best possible, so being the most accurate is optimizing for one outcome. Please clarify.

Figure 1: The goal of this figure is unclear. This looks like a graphical abstract but is inserted in the paper. The illustration of a patient standing with an IV bag is irrelevant to the study. The EUS does not present a report or interpretation. The charts are not readable in this figure.

Introduction:

Line 44 - This statement is misleading. Esophageal and gastric cancer staging has been thoroughly studied, including its relationship to survival in several papers. 

For example, see: 

Jürgensen, C., Brand, J., Nothnagel, M. *et al.*

 Prognostic relevance of gastric cancer staging by endoscopic ultrasound. *Surg Endosc*

 **27**, 1124–1129 (2013). https://doi.org/10.1007/s00464-012-2558-z

Brown, C., Howes, B., Jamieson, G.G. *et al.*

 Accuracy of PET-CT in Predicting Survival in Patients with Esophageal Cancer. *World J Surg*

 **36**, 1089–1095 (2012). https://doi.org/10.1007/s00268-012-1470-y

Chung, H.W., Lee, E.J., Cho, YH. *et al.*

 High FDG uptake in PET/CT predicts a worse prognosis in patients with metastatic gastric adenocarcinoma. *J Cancer Res Clin Oncol* **136**, 1929–1935 (2010). https://doi.org/10.1007/s00432-010-0852-5

Line 48: EUS is used in the abbreviated form before being introduced

Line 51: Check for double spaces

Materials and Methods:

Line 69: EMR is not defined before in the text. It is commonly used as an abbreviation for Electronic Medical Records, so this is especially unclear.

Line 72: NCCN is not defined

Line 74: Explain how you obtained or assumed data for patients you could not contact nor locate in the electronic medical records. Patients that were alive on follow-up were considered alive for the period, and patients with the date of passing in their medical records were believed to live until that date. What about patients that did not have any note regarding being deceased but did not have recent activity to indicate they were alive? How short was the interval for a patient to be considered alive?

Line 81: Having only patients without metastatic disease undergo EUS severely skews the results towards less severe cases. This hinders the comparison between PET-CT (which everyone got) and EUS, which is implied in the manuscript.

Line 91: State the number of years of experience each endoscopist had

Line 98: Check for double spaces

Line 103: Check for double spaces

Line 119: It is standard to use International System units for radiation dose (MBq)

Results:

Please avoid presenting redundant data in the text and tables. Include the most relevant points in the text and more detailed information in the tables.

Line 173: Include a breakdown of the schemes used and, if possible, perform a subgroup analysis indicating if one or more of them is associated with a change in prognosis following different staging by EUS or PET/CT.

Line 183: It is not described in the methods why you chose to compare T1-T2 vs. T3-T4. Please make this explicit. Moreover, it is known that EUS is especially useful in differentiating T3 from T4 tumors, so an individual T-stage analysis would be valuable to include.

Table 4: Please report the non-significant p-values as the p-value may hint at the effect size, sample heterogeneity, or lack thereof

Discussion:

Line 232: ESMO and NCCN guidelines mention EUS as the optimal strategy for local staging of esophageal and gastric cancers.

Line 256: Typo - "noy"

Line 294: Please mention in the limitations that only patients with metastatic disease ruled out by PET-CT proceeded to EUS, thereby reducing the complexity of the cases evaluated by it

Conclusion:

Please cite alternatives to the "suboptimal" methods you mention, such as PET/CT with more modern tracers, PET/MRI, etc.

Author Response

Thank you very much for allowing me to review this manuscript. The authors describe a retrospective study evaluating the performance of endoscopic ultrasound (EUS) and PET/CT for the restaging of esophageal, esophagogastric junction, and gastric cancer and correlate this restaging to survival metrics. The manuscript's strengths include a moderately large sample and sound statistical methods to determine survival. Weaknesses include the commonality of the approach, expected results, and suboptimal narrative structure. EUS is a mainstay of the locoregional staging of these cancers. FDG PET-CT has many limitations that are being overcome with the rise of new radiopharmaceuticals such as FAPI and other imaging technologies such as hybrid PET-MRI. Therefore, this manuscript, unfortunately, carries the burden of using out-of-date methods. 

Thank you for your comments, but I must respectfully disagree to some extent. Endoscopic ultrasound (EUS) is not considered a mainstay for preoperative staging. While it is commonly used for initial staging, there is ongoing discussion about its efficacy for restaging. Therefore, the primary objective of our work is to demonstrate its utility in this specific context. Our study's findings indeed support the performance of EUS in these patients, as it can potentially help avoid unnecessary laparoscopies and surgical interventions. Furthermore, we acknowledge your point that cutting-edge technologies such as FAPI or PET-MRI may hold promise. However, it is important to note that our study reflects real-life conditions, and in routine clinical practice, FDG-PET-CT remains a valuable tool. In fact, the National Comprehensive Cancer Network Guidelines (NCCN®) released in March 2023 recommend FDG-PET-CT for restaging, while EUS is not currently considered routine practice. This is where we believe our results bring significant novelty to the field. 

I raise below some points that could be better explained or improved:

Abstract:

Line 11 - EUS and PET-CT are used in the abbreviated form without a prior introduction

 Thanks. We have done so.

Line 24 - "EUS and PET-CT are suboptimal tools, but the most accurate, to determine esophagogastric cancer stage" - I would advise rephrasing this sentence as it can be deceiving. Optimal means the best possible, so being the most accurate is optimizing for one outcome. Please clarify.

Agreed. Thank you very much for this remark. We have changed this sentence, trying to make it more understandable.

 Figure 1: The goal of this figure is unclear. This looks like a graphical abstract but is inserted in the paper. The illustration of a patient standing with an IV bag is irrelevant to the study. The EUS does not present a report or interpretation. The charts are not readable in this figure.

This is a graphical abstract intended to represent a concept. The patient has been removed from the figure, leaving only an allegorical representation of chemotherapy and a small humanizing detail. The EUS picture has been relocated and clearly depicts a tumor. Including a figure legend in a graphical abstract would be unconventional. Additionally, the font size of the charts has been increased to ensure readability. 

Introduction:

Line 44 - This statement is misleading. Esophageal and gastric cancer staging has been thoroughly studied, including its relationship to survival in several papers. 

We understand your perspective. While there may be previous studies, such as the one mentioned in Surgical Endoscopy, with smaller populations and in patients who did not undergo preoperative chemotherapy, we acknowledge that we have not come across any studies that match the specific focus of our research. Our paper's primary contribution lies in investigating the role of EUS in restaging after neoadjuvant chemotherapy, an area that has not been extensively explored. By adding the term "EUS" to this paragraph, we aim to emphasize the originality of our research. We appreciate your valuable input and consideration. Thank you.

For example, see: 

Jürgensen, C., Brand, J., Nothnagel, M. *et al.*

 Prognostic relevance of gastric cancer staging by endoscopic ultrasound. *Surg Endosc*

 **27**, 1124–1129 (2013). https://doi.org/10.1007/s00464-012-2558-z

Brown, C., Howes, B., Jamieson, G.G. *et al.*

 Accuracy of PET-CT in Predicting Survival in Patients with Esophageal Cancer. *World J Surg*

 **36**, 1089–1095 (2012). https://doi.org/10.1007/s00268-012-1470-y

Chung, H.W., Lee, E.J., Cho, YH. *et al.*

 High FDG uptake in PET/CT predicts a worse prognosis in patients with metastatic gastric adenocarcinoma. *J Cancer Res Clin Oncol* **136**, 1929–1935 (2010). https://doi.org/10.1007/s00432-010-0852-5

Line 48: EUS is used in the abbreviated form before being introduced

 Thank you, we have changed this.

Line 51: Check for double spaces

 Thanks.

Materials and Methods:

Line 69: EMR is not defined before in the text. It is commonly used as an abbreviation for Electronic Medical Records, so this is especially unclear.

 Thanks, it endoscopic mucosal resection has been added there.

Line 72: NCCN is not defined

 Thanks, it has been defined.

Line 74: Explain how you obtained or assumed data for patients you could not contact nor locate in the electronic medical records. Patients that were alive on follow-up were considered alive for the period, and patients with the date of passing in their medical records were believed to live until that date. What about patients that did not have any note regarding being deceased but did not have recent activity to indicate they were alive? How short was the interval for a patient to be considered alive?

 The regional electronic medical record is highly detailed and accurate. We have added this paragraph that I think addresses your concerns.

The regional electronic medical system has integrated information from hospital and primary care settings. We can track patients from specialized and primary care with quite a close follow-up, with also information about death from the family physician and death certificates. In case of unclear information, we contacted directly with pa-tients or household relatives by telephone calls. The only patients lost to follow-up were virtually the ones who died.

Line 81: Having only patients without metastatic disease undergo EUS severely skews the results towards less severe cases. This hinders the comparison between PET-CT (which everyone got) and EUS, which is implied in the manuscript.

 We have slightly modified this paragraph. EUS was performed before and after chemotherapy. We only compared restaging and, indeed, the focuses on these cases, the ones who are neither the worse ones (metastatic) nor the milder ones (the ones which go directly to surgery). I think this small change clarifies this point.

Line 91: State the number of years of experience each endoscopist had

 They have been added.

Line 98: Check for double spaces

 Thanks.

Line 103: Check for double spaces

 Thanks

Line 119: It is standard to use International System units for radiation dose (MBq)

 It has been changed, leaving the mCI dosage inside brackets, as some nuclear medicine practitioners feel more comfortable with the mCI:

Results:

Please avoid presenting redundant data in the text and tables. Include the most relevant points in the text and more detailed information in the tables.

 We have modified the paper to some extent in this sense. The most reiterative subheading in the ‘tumor characteristic’ (lines 184-196), and it has been significantly shortened, deleting redundant information.

Line 173: Include a breakdown of the schemes used and, if possible, perform a subgroup analysis indicating if one or more of them is associated with a change in prognosis following different staging by EUS or PET/CT.

 The main schemes have been depicted, line 201-204. The logical reason for the presence of different schemes is the long period of time included. There were no signiffican differences with the different chemotherapy schemes. A statement has been conveniently added in line

Line 183: It is not described in the methods why you chose to compare T1-T2 vs. T3-T4. Please make this explicit. Moreover, it is known that EUS is especially useful in differentiating T3 from T4 tumors, so an individual T-stage analysis would be valuable to include.

 A statement has been included in the text (lines 155-158, page 5). We have performed the suggested comparison, and indeed, it is highly accurate. However, upon careful consideration, we have concluded that incorporating this information may introduce additional complexity and potentially make it less clear for the average reader. As it is not essential to the main focus of the paper, we have decided not to include this information at this time. However, we are open to your opinion and will proceed to include it if you believe it is of significant importance. We appreciate your thoughtful remarks and suggestions. Thank you.

Table 4: Please report the non-significant p-values as the p-value may hint at the effect size, sample heterogeneity, or lack thereof

 We have added p values to table 4

Discussion:

Line 232: ESMO and NCCN guidelines mention EUS as the optimal strategy for local staging of esophageal and gastric cancers.

 We have used preoperative as a synonym of restaging, and this can be misleading. For this reason, we have changed the term ‘preoperative’ for ‘restaging’ in this sentence. Thanks

Line 256: Typo - "noy"

 It has been corrected.

Line 294: Please mention in the limitations that only patients with metastatic disease ruled out by PET-CT proceeded to EUS, thereby reducing the complexity of the cases evaluated by it

 Well, this is the real-life practice. Metastatic patients should never go to EUS, because it adds nothing to their workup. Nevertheless, we have added a short statement in this sense.

Conclusion:

Please cite alternatives to the "suboptimal" methods you mention, such as PET/CT with more modern tracers, PET/MRI, etc.

We have added a statement.

An indepth revision of the English language has been performed.

Thank you very much for your insight. You have performed a thorough revision of the paper and we believe your comments and our reflection on them can improve the paper.

Reviewer 3 Report

I would like to congratulate the authors of an interesting article entitled “Preoperative EUS vs. PET-CT evaluation of response to neoadjuvant therapy for esophagogastric cancer and its correlation with survival”.

The manuscript is written clearly, in good quality English. The title is appropriate, the abstract contains all relevant information. In the introduction section, the authors summarize the current state of knowledge and give the rationale for the research. The methodology is written precisely, statistical analysis is correct, although its description is short. Results are presented clearly. In the discussion, the authors appropriately refer to the current literature. I have no major comments to the article. However, I think that the authors should introduce minor corrections.

1.       In the abstract, the authors use both “95%CI” and “CI95%”. I suggest you use one reporting format for example “95% CI:”. When it comes to statistical analysis and reporting of results, a very interesting article is “Statistical and data reporting guidelines for the European Journal of Cardio-Thoracic Surgery and the Interactive CardioVascular and Thoracic Surgery”, doi:10.1093/ejcts/ezv168.

2.       In visual abstract, “Charlson” instead of “Charlon” should be used.

3.       Pleas use “intravenous” instead of “iv”.

4.       The first time an abbreviation is used in the text, it should be explained in each case, this also applies to commonly known abbreviations such as CT (Line 52), PET-CT (Line 53), EUS (Line 53), EMR (Line 69).

5.       Please limit the use of uncommon abbreviations. Please consider using “distant metastases” instead of “M1”. “NT” is an abbreviation that is used very rarely. I believe that the use of “neoadjuvant therapy” or “induction therapy” would improve the clarity of the text period.

6.       Please consider explaining the abbreviations of neoadjuvant therapy “ECF, FLOT-4 or EOX” (Line 73), for example “ECF (epirubicin, cisplatin, fluorouracil), …”.

7.       Line 93 “EUS probe (Olympus, GF-UCT 165-AL5)” – manufacturer headquarters city and country should be stated “…GF-UCT 165-AL5 EUS probe (Olympus, Tokyo, Japan)”.

8.       Line 125: please specify “imaging technique”.

9.       In Table 4, pleas give the exact “p” values instead of “n.s.”

Overall, I found the article to be clear and very thorough, and of great clinical value. I suggest publishing the article in the Cancers after minor corrections.

Author Response

I would like to congratulate the authors of an interesting article entitled “Preoperative EUS vs. PET-CT evaluation of response to neoadjuvant therapy for esophagogastric cancer and its correlation with survival”.

Thank you very much for this encouraging general assessment.

The manuscript is written clearly, in good quality English. The title is appropriate, the abstract contains all relevant information. In the introduction section, the authors summarize the current state of knowledge and give the rationale for the research. The methodology is written precisely, statistical analysis is correct, although its description is short. Results are presented clearly. In the discussion, the authors appropriately refer to the current literature. I have no major comments to the article. However, I think that the authors should introduce minor corrections.

  1. In the abstract, the authors use both “95%CI” and “CI95%”. I suggest you use one reporting format for example “95% CI:”. When it comes to statistical analysis and reporting of results, a very interesting article is “Statistical and data reporting guidelines for the European Journal of Cardio-Thoracic Surgery and the Interactive CardioVascular and Thoracic Surgery”, doi:10.1093/ejcts/ezv168.

Thanks for the reference. We have downloaded and check it. ‘95% CI’ has been introduced in every confidence internval in the paper.

  1. In visual abstract, “Charlson” instead of “Charlon” should be used.

It has been corrected.

  1. Pleas use “intravenous” instead of “iv”.

This has also been changed as suggested.

  1. The first time an abbreviation is used in the text, it should be explained in each case, this also applies to commonly known abbreviations such as CT (Line 52), PET-CT (Line 53), EUS (Line 53), EMR (Line 69).

We have included the full term in their first appearance in the main text and the abstract.

  1. Please limit the use of uncommon abbreviations. Please consider using “distant metastases” instead of “M1”. “NT” is an abbreviation that is used very rarely. I believe that the use of “neoadjuvant therapy” or “induction therapy” would improve the clarity of the text period.

  1. Please consider explaining the abbreviations of neoadjuvant therapy “ECF, FLOT-4 or EOX” (Line 73), for example “ECF (epirubicin, cisplatin, fluorouracil), …”.

They have been conveniently explained in the first time they appear in the text.

  1. Line 93 “EUS probe (Olympus, GF-UCT 165-AL5)” – manufacturer headquarters city and country should be stated “…GF-UCT 165-AL5 EUS probe (Olympus, Tokyo, Japan)”.

This has been added. Thanks.

  1. Line 125: please specify “imaging technique”.

It has been conveniently specified.

Reviewer 4 Report

This paper's study has great clinical value, which regards EUS and PET-C role on evaluation of esophagogastric cancer and its correlation with survival. However,  I have the following questions which need to answer.

(1) What's the abbreviation of MER in line 69? 

(2) What's the lymph nodes positive standard for PET-CT, because there is a  big difference of positive lymph nodes number between PET-CT and EUS. Please add it, which is very important.

(3)In line 201, there is no period. Please check other errors, for example ”95% CI:", "% [IC 95]" or % [CI 95]"(Table 2), paragraph format, et al.

(4) In line 240, "PET" should be "PET-CT", because PET and PET-CT are different. 

(5) In the restaging part of table 1, there is no T1 for Depth of invasion. Please add it whatever it is.

(6) In line 256,  should "noy" be "not"?

(7) In figure 2, what is the p value?

(8) What is the PET-CT scanning time after neoadjuvant therapy? Please add it, because It is not advocated to have PET-CT exam immediately after neoadjuvant therapy, which will affect the FDG uptake due to inflammatory response.

(9) In the last paragraph of discussion part, it is said "Our study has some limitations. First",  where is second?

Author Response

This paper's study has great clinical value, which regards EUS and PET-C role on evaluation of esophagogastric cancer and its correlation with survival. However,  I have the following questions which need to answer.

  • What's the abbreviation of MER in line 69? 

Endoscopic mucosal resection. It has been added to the text.

  • What's the lymph nodes positive standard for PET-CT, because there is a  big difference of positive lymph nodes number between PET-CT and EUS. Please add it, which is very important.

As usual, they were considered positive when the FDG uptake was higher than the liver. A statement has been added in lines 151-152. We agree this is important to depict.

(3)In line 201, there is no period. Please check other errors, for example ”95% CI:", "% [IC 95]" or % [CI 95]"(Table 2), paragraph format, et al.

This has been corrected. Thank you.

  • In line 240, "PET" should be "PET-CT", because PET and PET-CT are different. 

Thank you. We have found the same mistakes in other places in the paper, and it has been corrected.

  • In the restaging part of table 1, there is no T1 for Depth of invasion. Please add it whatever it is.

Not even one in restaging, but it has been added to the table as you request.

  • In line 256,  should "noy" be "not"?

We have corrected this.

  • In figure 2, what is the p value?

It has been added to the figure.

  • What is the PET-CT scanning time after neoadjuvant therapy? Please add it, because It is not advocated to have PET-CT exam immediately after neoadjuvant therapy, which will affect the FDG uptake due to inflammatory response.

It was performed two weeks after completing neoadjuvant chemotherapy. This has been added to lines 85-86 of the paper.

  • In the last paragraph of discussion part, it is said "Our study has some limitations. First",  where is second?

The second has been added. The fact that metastatic patients did not undergo EUS assessment, therefore selecting fewer complex cases. Is has been added to the text.

We thank you very much your constructive approach and your excellent insights.

Round 2

Reviewer 2 Report

Dear Authors,

Thank you very much for allowing me to review this manuscript. The authors have performed an extensive review and editing process that led to significant improvements over the last version submitted. Most of my concerns have been sufficiently addressed. My only additional suggestion would be to include the individual T-stage analysis to compare the discriminatory power of EUS and PET/CT that was performed but omitted. I agree with the authors that this is not the focus of the paper, but including these results would be beneficial to the community and to the journal as they could be used for further research including review papers and meta-analyses. Therefore, I would suggest including the analysis, at least as a table (including the number of cases in each category vs the ground truth if the intent is to allow for later meta-analyses).

I appreciate the authors' efforts in revising the manuscript and congratulate them for their great work.

Author Response

Dear Authors,

Thank you very much for allowing me to review this manuscript. The authors have performed an extensive review and editing process that led to significant improvements over the last version submitted. Most of my concerns have been sufficiently addressed. My only additional suggestion would be to include the individual T-stage analysis to compare the discriminatory power of EUS and PET/CT that was performed but omitted. I agree with the authors that this is not the focus of the paper, but including these results would be beneficial to the community and to the journal as they could be used for further research including review papers and meta-analyses. Therefore, I would suggest including the analysis, at least as a table (including the number of cases in each category vs the ground truth if the intent is to allow for later meta-analyses).

Thanks. EUS accuracy in the discrimination of T and N stages is stated in the text, lines 236-241. For PET-CT we do not have T staging as nuclear medicine practitioners state T staging hardly ever, depicting just tumor location, SUV, N positive lymph nodes and the presence of metastases. Absence of a precise PET-CT T staging was stated originally in the methods (line 158-161). T and N staging has been depicted in table 2, with absolute number that can be eventually used in a metanalysis in the future. For this purpose, we have also offered to share our dataset to any given researcher with interest in further analysis (line 391).

We agree with you about the interest on this issue, and so it has been added for EUS.

Thank you very much for your interest in the paper.  

I appreciate the authors' efforts in revising the manuscript and congratulate them for their great work.

Reviewer 4 Report

Now the revised version has a great improvement. But there still be some minor errors or questions needed to answer.

(1) In line 11, should "E Endoscopic" be "Endoscopic"?

(2) In line 218, is "un PET" supposed to be "on PET"? 

(3) The authors replied that "As usual, they were considered positive when the FDG uptake was higher than the liver. A statement has been added in lines 151-152", which is still not showed in paper. And it's not good enough for positive lymph nodes standard, which also should include size, shape, SUVmax value. That's the reason that there is a big difference of positive lymph nodes between PET-CT and EUS. Please add it.

Author Response

Now the revised version has a great improvement. But there still be some minor errors or questions needed to answer.

(1) In line 11, should "E Endoscopic" be "Endoscopic"?

(2) In line 218, is "un PET" supposed to be "on PET"? 

Thanks for your accurate revision. Both mistakes have been fixed.

(3) The authors replied that "As usual, they were considered positive when the FDG uptake was higher than the liver. A statement has been added in lines 151-152", which is still not showed in paper. And it's not good enough for positive lymph nodes standard, which also should include size, shape, SUVmax value. That's the reason that there is a big difference of positive lymph nodes between PET-CT and EUS. Please add it.

Sorry, the statement is in lines 159-161. After your suggestion, we have addressed our nuclear medicine colleagues who consider this statement more adequate: “Lymph nodes were considered positive for metastasis when there was 18F-FDG uptake higher than that in the liver. Other factors such as lymph node short axis and SUVmax were also considered when studying PET/CT N staging.”

Indeed, a SUVmax cutoff threshold has not been established, and size is controversial, not just only for PET/CT but also for EUS.

Thank you very much for your careful revision.